Taxonomic revision of the threatened African genus Pseudohydrosme Engl. (Araceae), with P. ebo, a new, critically endangered species from Ebo, Cameroon

http://orcid.org/0000-0003-4343-3124 Cheek Martin 1 m.cheek@kew.org
http://orcid.org/0000-0002-7319-9108 Tchiengué Barthélemy 2
van der Burgt Xander 3
1 Royal Botanic Gardens, Kew , Richmond , UK
2 Institute of Agronomic Research and Development, Herbier National Camerounais , Yaoundé, Centrale , Cameroon
3 Identification & Naming, Royal Botanic Gardens, Kew , Richmond, Surrey , UK
Sosa Victoria
Electronic publication date: 2021 Feb 11
Publication date: 2021
Volume: 9
Electronic Location ID: e10689
Received 2020 Oct 13; Accepted 2020 Dec 11
Copyright: © 2021 Cheek et al.
Copyright year: 2021
Copyright holder: Cheek et al.
License: This is an open access article distributed under the terms of the Creative Commons Attribution License, which permits unrestricted use, distribution, reproduction and adaptation in any medium and for any purpose provided that it is properly attributed. For attribution, the original author(s), title, publication source (PeerJ) and either DOI or URL of the article must be cited.
License URL: https://creativecommons.org/licenses/by/4.0/

Keywords: Apomixis, Cross-Sanaga interval, Disjunction, Extinction, Ornamental, Vivipary

Funding: Garfield Weston Foundation and the Bentham Moxon Trust Cameroon Tropical Important Plant Areas programme Fieldwork in Cameroon was supported by the Garfield Weston Foundation and the Bentham Moxon Trust. Players of the People’s Postcode Lottery support the Cameroon Tropical Important Plant Areas programme for which this paper is an output. The funders had no role in study design, data collection and analysis, decision to publish, or preparation of the manuscript.

==============================
This is the first revision in more than 100 years of the African genus Pseudohydrosme, formerly considered endemic to Gabon. Closely related to Anchomanes, Pseudohydrosme is distinct from Anchomanes because of its 2-3-locular ovary (vs. unilocular), peduncle concealed by cataphylls at anthesis and far shorter than the spathe (vs. exposed, far exceeding the spathe), stipitate fruits and viviparous (asexually reproductive) roots (vs. sessile, roots non-viviparous), lack of laticifers (vs. laticifers present) and differences in spadix: spathe proportions and presentation. However, it is possible that a well sampled molecular phylogenetic analysis might show that one of these genera is nested inside the other. In this case the synonymisation of Pseudohydrosme will be required. Three species, one new to science, are recognised, in two sections. Although doubt has previously been cast on the value of recognising Pseudohydrosme buettneri, of Gabon, it is here accepted and maintained as a distinct species in the monotypic section, Zyganthera. However, it is considered to be probably globally extinct. Pseudohydrosme gabunensis, type species of the genus, also Gabonese but probably extending to Congo, is maintained in Sect. Pseudohydrosme together with Pseudohydrosme ebo sp.nov. of the Ebo Forest, Littoral Region, Cameroon, the first addition to the genus since the nineteenth century, and which extends the range of the genus 450 km north from Gabon, into the Cross-Sanaga biogeographic area. The discovery of Pseudohydrosme ebo resulted from a series of surveys for conservation management in Cameroon, and triggered this article. All three species are morphologically characterised, their habitat and biogeography discussed, and their extinction risks are respectively assessed as Critically Endangered (Possibly Extinct), Endangered and Critically Endangered using the IUCN standard. Clearance of forest habitat for logging, followed by agriculture or urbanisation are major threats. Pseudohydrosme gabunensis may occur in a formally protected area and is also cultivated widely but infrequently in Europe, Australia and the USA for its spectacular inflorescences.

Introduction

The new species reported in this monograph was discovered as a result of a long-running survey of plants in Cameroon to support improved conservation management. The survey is led by botanists from the Royal Botanic Gardens, Kew and the National Herbarium of Cameroon-IRAD (Institute for Research in Agronomic Development), Yaoundé. This study has focussed on the Cross-Sanaga interval (Cheek et al., 2001) which contains the area with the highest species diversity per degree square in tropical Africa (Barthlott, Lauer & Placke, 1996). The herbarium specimens collected in these surveys formed the foundations for a series of Conservation Checklists (see below). So far, they have resulted in the discovery and publication of over 100 new species and several new genera, the recognition of new protected areas and the production of scientific data for the Cameroon Important Plant Area programme (https://www.kew.org/science/our-science/projects/tropical-important-plant-areas-cameroon).

In October 2015 the last two authors found two leafless, flowering plants of a spectacular aroid in the Ebo Forest of Littoral Region, Cameroon (van der Burgt 1888, Fig. 1). Since these had prickles on the peduncle they were provisionally identified as Anchomanes Schott.

Figure 1 Pseudohydrosme ebo (van der Burgt 1888, K, YA).

Photo of flowering, leafless plant at Ebo forest, October 2015. Photo by Xander van der Burgt.

In late 2018 the herbarium specimen of this collection was redetermined by the first author as Pseudohydrosme Engl., an erstwhile Gabonese genus previously unknown from Cameroon. Pseudohydrosme is distinguished from Anchomanes by a peduncle much shorter than the spathe (vs.far longer) and by 2–3-locular (vs.unilocular) ovaries (Mayo, Bogner & Boyce, 1997). van der Burgt 1888 was suspected of representing a species new to science since it differed in several characters from the two known species of Pseudohydrosme. In addition, Ebo in Cameroon is geographically 450 km distant from the range of those two previously known species in Gabon, and is in a different biogeographic zone. In order to obtain the missing stages of fruit and leaf to complement van der Burgt 1888, it was decided to revisit the collection site at the next available opportunity. Hence, in December 2019 leaves, although not fruits, and additional field data were obtained (van der Burgt 2377, Fig. 2) including from a further site. Additional characters separating the Ebo taxon from other members of the genus were discovered in the newly collected material. Early in 2020 a previously overlooked flowering specimen, Morgan 25 came to light.

Figure 2 Pseudohydrosme ebo (van der Burgt 2377, K, YA).

Photo of plant in leaf, in habitat, Ebo forest, December, 2019. Photo by Xander van der Burgt.

Pseudohydrosme was described by A. Engler with two species, P. gabunensis Engl. based on Soyaux 299 (B) collected 13 October 1881 and P. buettneri Engl. based on Büttner (Buettner) 519 (B) collected in September 1884, both from forest at Sibang, formerly near and now largely subsumed by, Libreville, the capital and principal city of Gabon (Engler, 1892; Bogner, 1981).

In 1973 the renowned aroid specialist Josef Bogner visited Gabon and rediscovered two plants of P. gabunensis. Tubers were taken to Germany and cultivated allowing description of the leaves of Pseudohydrosme for the first time (Bogner, 1981).

The second species of the genus, P. buettneri has never been refound. It differs so greatly from the first in the structure of its inflorescence (see key and species account below) that the noted aroid specialist N.E. Brown erected a separate genus, Zyganthera N.E. Brown for it (Thistleton-Dyer, 1901). Engler (1911) reduced Zyganthera to sectional rank within Pseudohydrosme.

In their monumental account ‘The Genera of Araceae’, Mayo, Bogner & Boyce (1997) placed Nephthytis Schott as sister to Pseudohydrosme + Anchomanes, in the tribe Nephthytideae Engl. Molecular phylogenetic analysis has subsequently supported the close relationship of Pseudohydrosme and Anchomanes, but since each were represented by only a single taxon, it was not possible to test their monophyly or sister relationship (Cabrera et al., 2008; Cusimano et al., 2011; Nauheimer, Metzler & Renner, 2012). In presenting new data on Pseudohydrosme gabunensis based on successful pollination of flowers and fruit production on plants in cultivation in the Netherlands, Hetterscheid & Bogner (2013) questioned the distinction of Pseudohydrosme and Anchomanes. They considered the only difference to be the locularity of the ovaries (2–3 vs. 1) and set aside the difference in peduncle: spathe proportions maintained in Mayo, Bogner & Boyce (1997). However, later in their article Hetterscheid & Bogner (2013) then brought to light two new characters that further support the distinction of Pseudohydrosme from both Anchomanes and from all other aroids (see “Discussion” below). Hetterscheid & Bogner (2013) made the case to extend the range of Pseudohydrosme gabunensis from Gabon southwards into the Republic of Congo (Congo-Brazzaville) citing photos and a specimen deposited at WAG by Ralf Becker. However, we have not been able to access the material in order to verify this statement (see “Methods”). Despite this, we accept this record since it is confirmed by Alistair Hay (in litt.) who has seen it in flower and confirms that it conforms to Pseudohydrosme gabunensis in spathe shape and colour.

In this article we describe the Cameroon material from Ebo Forest as a new species to science, Pseudohydrosme ebo Cheek, in the context of a revision of the genus, last revised over 100 years ago (Engler, 1911).

Materials and Methods

The electronic version of this article in Portable Document Format (PDF) will represent a published work according to the International Code of Nomenclature for algae, fungi and plants (ICN), and hence the new names contained in the electronic version are effectively published under that Code from the electronic edition alone. In addition, new names contained in this work which have been issued with identifiers by IPNI will eventually be made available to the Global Names Index. The IPNI LSIDs can be resolved and the associated information viewed through any standard web browser by appending the LSID contained in this publication to the prefix “http://ipni.org/”. The online version of this work is archived and available from the following digital repositories: PeerJ, PubMed Central and CLOCKSS.

The fieldwork in Cameroon was conducted under the terms of the Memorandum of Collaboration between IRAD-Herbier National de Cameroun and Royal Botanic Gardens, Kew that is valid until 5 September 2021. The research permit number for the last author in 2019 was 000146/MINRESI/B00/C00/C10/C12 (issued 28 November 2019), and the export permit number was 098/IRAD/DG/CRRA-NK/SSRB/12/2019 (issued 19 December 2019). In 2015 the respective numbers were: 00076/MINRESI/B00/C00/C10/C14 (issued 25 August 2015) and 103/IRAD/DG/CRRA-NK/SSRB-HN/10/2015 (issued 23 October 2015). Fieldwork was approved by the Institutional Review Board of the Royal Botanic Gardens, Kew entitled the Overseas Fieldwork Committee (OFC) under the registration numbers OFC 673-1 (2015) and OFC 807-3 (2019). The most complete set of duplicates for all specimens made was deposited at YA, the remainder exported to K for identification and distribution following standard practice (Cheek & Cable, 1997).

Herbarium citations follow Index Herbariorum (Thiers, 2020). Specimens were viewed at B, BR, K, P, WAG and YA. Pseudohydrosme is centred in Gabon. The national herbarium of Gabon is LBV, but the most comprehensive herbaria for herbarium specimens of that country are P and WAG. The National Herbarium of Cameroon, YA, was also searched for additional material but without success. During the time that this article was researched, it was not possible to obtain physical access to material at WAG (due to the transfer of WAG to Naturalis, Leiden, subsequent construction work and covid-19 travel and access restrictions). However, images for WAG specimens were studied at https://bioportal.naturalis.nl/?language=en and those from P at https://science.mnhn.fr/institution/mnhn/collection/p/item/search/form?lang=en_US. We also searched JStor Global Plants (2020) for additional type material of the genus, and finally the Global Biodiversity Facility (GBIF, www.gbif.org accessed 23 August 2020) which lists 28 occurrences and 21 images, mainly relating to the holdings (including duplicate herbarium sheets) of WAG, followed by P.

Binomial authorities follow the International Plant Names Index (IPNI, 2020). The conservation assessment was made using the categories and criteria of IUCN (2012). GeoCAT was used to calculate Red List metrics (Bachman et al., 2011). Spirit preserved material was not available. Herbarium material was examined with a Leica Wild M8 dissecting binocular microscope fitted with an eyepiece graticule measuring in units of 0.025 mm at maximum magnification. The drawings were made with the same equipment using Leica 308700 camera lucida attachment. This was used to characterise and measure in particular features of the flowers. The herbarium specimens of the new species described below as Pseudohydrosme ebo were soaked in warm water to enable the spathe to be folded back, exposing the spadix and hydrated flowers. The terms and format of the description follow the conventions of Mayo, Bogner & Boyce (1997). Georeferences for specimens lacking latitude and longitude were obtained using Google Earth (https://www.google.com/intl/en_uk/earth/versions/). The map was made using SimpleMappr (https://www.simplemappr.net).

Results

Taxonomic treatment

Pseudohydrosme is closely related to Anchomanes Schott (7–8 species) and the pair are in a sister relationship with Nephthytis Schott (six species). Together these three genera are all tropical African (all are absent from Madagascar). However, anomalous Nephthytis bintuluensis A. Hay, J. Bogner and P. C. Boyce occurs in Borneo (Hay, Bogner & Boyce, 1994; Nauheimer, Metzler & Renner, 2012). These three genera comprise the Nephthytideae which is sister to the SE Asian tribe Aglaonemateae Engl. (Cabrera et al., 2008), consisting of Aglaonema Schott (Van, Nguyen-Phi & Luu, 2020). Both groups share adjacent male and female flower zones, free stamens and collenchyma arranged in threads peripheral to the vascular strands of leaf blades and petioles (with the exception of Nephthytis, in which collenchyma can form interrupted bands (Keating, 2002; Cabrera et al., 2008).

Morphological support for Nephthytideae is the well-developed posterior costa, that is, ± tripartite primary development (Cusimano et al., 2011: Table 1)

Table 1 Characters separating Pseudohydrosme gabunensis from Pseudohydrosme ebo and Pseudohydrosme buettneri.

	Pseudohydrosme gabunensis	Pseudohydrosme ebo	Pseudohydrosme buettneri	
Leaf lobe posture and surface	Pendulous, surface bullate (convex between nerves)	Horizontal, surface flat	Unknown	
Spadix male: female zone width	Base of male zone abruptly wider than female zone below	Male zone about as wide as female zone	Male zone about as wide as female zone	
Spathe length (cm)	(30–)40–55(–70)	24–30(–34.5)	80	
Spathe tube: colour of outer surface	Uniformly yellow (or white)	Vertical broad pink-brown stripes on white background	Unknown	
Spathe blade: colour of inner surface	Yellow or white around margins, central area dark red-purple. Demarcation abrupt	Light reddish brown to pink with pale green veins, gradually slightly darkening in the central area	Unknown	
Number of ovary locules and stigma lobes	2 (–3)	(2–) 3	2	
Female flower density	Covering completely the female zone spadix axis	Lax (sparse), spadix axis visible between the female flowers	Covering completely the female zone spadix axis	
Female and male flower zone contiguity	Female and male zones contiguous, male and female flowers in contact with each other	Female zone separated from male zone by a short naked portion of spadix axis	Female and male zones contiguous, male and female flowers in contact with each other	
Notes:

Data for Pseudohydrosme gabunensis from Bogner (1981:33), Hetterscheid & Bogner (2013:104–113), Hay (in litt.), and live material cultivated at Royal Botanic Gardens, Kew.

Dracontioid leaf divisions characterise Pseudohydrosme and Anchomanes but are not present in Nephthytis. They derive from complex splitting of a ‘simple’ and virtually entire blade as the leaf unfurls after emergence, which is why the leaflets at the margin are truncate with two ‘tips’. This is unique to Pseudohydrosme/Anchomanes (Hay, 2019: 285). Differences between Pseudohydrosme and Anchomanes are re-assessed in the discussion below.

Pseudohydrosme Engl. (Engler, 1892: 455; Brown in Thistleton-Dyer, 1901: 160; Engler, 1911: 47; Mayo, Bogner & Boyce, 1997: 221–222)

Type species: Pseudohydrosme gabunensis Engl. (Lectotypified by N. E. Brown in Thistleton-Dyer, 1901: 160).

Zyganthera N. E. Br. (Brown in Thistleton-Dyer, 1901:160). Heterotypic synonym.

Type and only species: Pseudohydrosme buettneri Engl. (Engler, 1892).

Large, seasonally dormant, monoecious herbs. Rhizome shallowly subterranean, the growing point at ground level, subglobose or cylindrical with annular leaf scars, and erect to horizontal or obliquely inclined, growing continuously and not renewed with each growing period. Roots fleshy, produced along length of rhizome, sometimes reproductive (the distal ends rising to the surface and producing new plants). Leaf solitary, large, emerging from several cataphylls; petiole cylindrical, erect, long, with minute and sparse prickles, sheath very short and inconspicuous. Blade transitioning from simple, sagittate and entire in seedlings, leaves of older plants developing slits and divisions (see above), in mature plants leaves dracontoid : trisect, primary divisions pinnatisect, distal lobes mostly truncate and bifid, sessile and decurrent, proximal lobes acuminate; primary lateral veins of ultimate lobes pinnate, often forming a regular submarginal collective vein (P. ebo) or an irregular collective vein, or veins running into margin (often in P. gabunensis), higher order venation reticulate.

Inflorescence solitary, appearing separately from the leaf. Cataphylls papery-membranous, (3–)4–6, proximal ± triangular, small, distal oblong-elliptic, exceeding spathe tube, pink-brown or red-brown, sometimes (P. gabunensis) spotted white. Peduncle concealed by cataphylls at anthesis, terete, very short <1/10th the length of the spathe, with minute, sparse, prickles. Spathe large, fornicate, unconstricted/very broad, with flaring auriculate margins; tube convolute, fleshy, obconic, with a few sparse prickles on the outer surface proximally. Spadix short, about 1/10–1/4 length of spathe, sessile, female zone subcylindric (P. ebo) obconic (P. buettneri) or gently obconic or conic (P. gabunensis), male zone cylindric, obtuse or rounded, subequal to ± twice (±four times in P. buettneri) as long as female, completely covered in flowers and fertile to apex (P. gabunensis) or with a distal appendix twice as long as the fertile portion and covered in sterile male flowers (P. buettneri) or flowers only laxly covering the spadix axis in the female zone and with the distal part of the axis of the female zone completely naked in places (P. ebo).

Flowers unisexual, perigone absent. Male flower 2–5-androus, stamens free, subprismatic, compressed, anthers sessile, connective thick, broad, overtopping thecae, thecae oblong, long, lateral, dehiscing by apical pores. Pollen extruded in strands, inaperturate, ellipsoid-oblong, very large (mean 106 micrometres diam.) exine psilate to slightly scabrous. Sterile male flowers (P. buettneri) composed of subprismatic, free staminodes. Female flower ovary globose to broadly ellipsoid, usually prismatic, 2–3-locular, ovules 1 per locule, anatropous, funicle short, placentation axile, at base of septum, stylar region attenuate to cylindric, narrower than ovary, stigma thick, shallowly 2–3-lobed or subdiscoid, concave centrally, wet when receptive.

Infructescence with slightly accrescent peduncle. Berry at first white, ripening dark purple, fleshy, wrinkled when mature, oblong-ellipsoid, laterally compressed to slightly bilobed, stipitate, large; stigma and style persistent (known only in P. gabunensis). Seeds subglobose to broadly ovoid, one side convex, the other slightly flattened, testa thin, whitish, smooth, papery, transparent; embryo large, outer surface green, inner white, endosperm absent, raphe distinct, hilum and micropyle purple, plumule with leaf primordia. Three species.

This description is based on that of Mayo, Bogner & Boyce (1997), with the addition of descriptions of the fruit, seed and roots of P. gabunensis, mainly from (Hetterscheid & Bogner, 2013), with novel features of the nervation, spathe and spadix from P. ebo (below in this article)

Phenology: flowering September and October (or March in cultivation in Europe); in leaf Dec.-April.

Distribution and habitat: Cameroon, Gabon and Congo(Brazzaville), lowland evergreen forest on coastal sediments (Gabon) or inland foothills on basement complex rocks (Cameroon) (Fig. 3).

Figure 3 Global distribution of the species of the African genus Pseudohydrosme.

Red dot = P. buettneri; black dots= P.gabunensis (location not available for Congo specimen); blue dot = P. ebo.

Etymology: meaning “false Hydrosme”, Hydrosme Schott is a synonym of Amorphophallus.

Local name and uses: none are documented.

Conservation: all species are highly infrequent and globally threatened according to IUCN (2012) criteria (see species accounts below), and P. buettneri is possibly extinct (not seen for over 100 years, the majority of its former habitat destroyed).

Pollination in the wild has not been investigated in detail in Pseudohydrosme, but is almost certainly by insects as is usual in Araceae. Two different species of flies, and two of beetles were reported to visit P. gabunensis (see below). In cultivation the stigmas are reported to be wet and receptive for only two days, and the scent reported to be faint, of lettuce (Lactuca) in the same species (see below also). Following successful fertilisation, seed development is reported to take up to 10 months in P. gabunensis (see below). Seed dispersal is probably by either ground-dwelling mammals or birds consuming the thinly fleshy purple berries.

DNA analysis was performed by Cabrera et al. (2008) for Pseudohydrosme gabunensis, using five regions of coding (rbcL, matK) and noncoding plastid DNA (partial trnK intron, trnL intron, trnL–trnF spacer). These sequences were subsequently used by Cusimano et al. (2011) and Nauheimer, Metzler & Renner (2012). The voucher is Wieringa 3308 (WAG), identified by Hetterschied, GenBank codes are AM905760, AM920582, AM932319 + AM933315.

Cultivation of one species, Pseudohydrosme gabunensis is unknown in Africa yet widespread but infrequent in the tropical glass-house collections of several large extra-African botanical gardens, mainly in Europe, Australia and N. America (see under that species).

Chromosome numbers are reported of one species, Pseudohydrosme gabunensis, as 2n = ca. 40 (Mayo, Bogner & Boyce, 1997; Bogner & Petersen, 2007).

Germination in Pseudohydrosme gabunensis is cryptocotylar and takes 3 weeks to 10 months. The large seed embryo remains buried, producing a single hastate seedling leaf (Hetterscheid & Bogner, 2013). The seedling type is C2 in the classification of Tillich (2014).

Medicinal uses, and chemistry is unreported in Pseudohydrosme. However, the much more frequent sister genus Anchomanes, is harvested as a traditional medicine for example in Cameroon, and contains bioactive compounds (Cheek, 1992)

Identification key to the sections and species of Pseudohydrosme

1. Spadix with distal half covered in sterile male flowers. Sect. Zyganthera1. P. buettneri

1. Spadix lacking sterile flowers, distal part with fertile male flowers only.Sect. Pseudohydrosme…2

2. Male and female zones of spadix contiguous; entire spadix covered in flowers densely arranged in both male and female zones; spathe blade inner surface yellow, greenish yellow or white with abrupt transition to a central dark red area;

stigmas 2(–3)-lobed. Gabon (probably Congo-Brazzaville)2. P. gabunensis

2. Male and female areas of spadix incompletely contiguous; female flowers laxly arranged with axis of female zone partly naked especially distally; spathe blade inner surface light reddish brown or pink, with wide green veins, very gradually becoming darker towards the centre; stigmas 3(–2)-lobed. Cameroon3. P. ebo

Sect. Zyganthera (N. E. Br.) Engl. (Engler, 1911: 49)

Type (and only) species: Pseudohydrosme buettneri Engl.

Zyganthera N. E. Br. (Brown in Thistleton-Dyer, 1901:160). Basionym

Male flowers with stamens in pairs; distal half of spadix covered in sterile male flowers; ratio of female:male (including sterile male) spadix portions c.1:4

1. Pseudohydrosme buettneri Engl. (Engler, 1892:456; Engler & Prantl, 1897: 59; Brown in Thistleton-Dyer, 1901: 160; Engler, 1911: 49).—Figs. 3 and 4.

Figure 4 Pseudohydrosme buettneri (Buettner 519, B).

Drawing of flowering plant. (A) Habit, rhizome and inflorescences; (B) spadix after removal of spathe showing from base to apex, female, male and staminodal flowers; (C and D) paired stamens, side view; (E) paired stamens flowers, transverse section; (F and G) staminodes, sterile male flower; (H) female flower, entire, side view; (J) female flower, longitudinal section. Reproduced from protologue, (Engler, 1892: taf. XVII). Drawn by Josef Pohl.

Holotype: Gabon, Estuaire Province, Libreville “Gabun, Mundagebiet; Sibange-Farm” fl. Sept. 1884, Buettner 519 (Holotype B destroyed or mislaid).

Terrestrial herb, rhizome vertical, subglobose, 2.5 cm long, 2.5 cm wide, surface tuberculate, roots fleshy, from along the length of the rhizome. Leaf unknown.

Inflorescence: Cataphylls three or more, 2–13 × 0.9–1 cm; peduncle 3 cm long, colour and indumentum unknown. Spathe 80 cm long, within pale except for the median longitudinal part which is dark purple. Spadix subcylindrical, 7–8 cm long, c. 1.3 cm diam. Female zone obconic 1.3 cm long. Fertile male zone 2 cm long. Appendix (of sterile male flowers) 5 cm long, c. 1.5 cm diam.

Female flowers 4 mm long, ovary globose-ovoid, 3 mm diam., style 1 mm long, slender; stigma bilobed, 1 mm diam., thick; ovules shortly ovoid, solitary in each locule and attached near the base of the septum. Male flowers with stamens 2 mm long, 2 mm wide, usually the two stamens of a flower, appressed to one another, with bilocular subextrorse thecae nearly reaching the stamen apex. Staminodes subprismatic, 4–6-sided, lacking anther thecae and much smaller in diameter than the stamens (description taken from Engler, 1892: 456 and tab. XV).

Phenology: flowering in September.

Local name and uses: none are known.

Etymology: named for the collector of the only known, and type specimen, Oscar Alexander Richard Buettner (Büttner)—(1858–1927), traveller and collector.

Distribution and ecology: known only from Sibang in Libreville, coastal lowland evergreen forest dominated by Aucoumea klaineana Pierre.

Additional specimens: none are known.

Notes: Pseudohydrosme buettneri has the largest inflorescence by far of all known species of the genus, with an 80 cm long spathe (however, 70 cm has been reached for P. gabunensis in cultivation in Australia according to Hay in litt.). The type specimen had lost the top part of the spathe, but dimensions were given by the collector (Engler, 1892).

The type, and only known specimen was at B, but is reported to be no longer there (Bogner, 1981). It may have been lost in the allied bombing of Berlin in March 1943, when most of the specimens at B were destroyed. However, the type specimen of P. gabunensis (see below) dating from about the same time, and also housed at B, has survived.

No additional specimens of this species have been found in the 136 years ensuing from collection of the type specimen. Hetterscheid & Bogner (2013) have questioned whether this species is not just a variant of P. gabunensis. However, this seems highly unlikely, because the specimen differs in three independent characters from P. gabunensis (and P. ebo):

the ratio between the female zone and the male zone (of fertile and sterile flowers) differs greatly between the two. In P. buettneri the ratio is 1:4+, while in the other two species it is less than 1:2.

in P. buettneri most of the spadix consists of an appendix of sterile male flowers. No such sterile appendix occurs in the other two species. In fact this character is otherwise unknown in the entire Aglaonemateae/Nephthytideae clade

in P. buettneri the stamens are paired (Engler, 1892), while in the other two species the stamens are not reliably paired, but also present in an indistinct ring of three to five.

Additional differences between the species can be seen in the pistil and spadix. The style in P. buettneri is <1/4 the width of the ovary. In P. gabunensis it is 1/2. The spadix of P. buettneri is cylindrical, and unconstricted, while that of P. gabunensis shows a pronounced constriction at the junction of female and male zones, and the male zone attains a greater width than the female zone.

Conservation. Pseudohydrosme buettneri is here assessed as Critically Endangered (Possibly Extinct). This is because it has only been found once, at a single site, in the “Munda region” at Sibang Farm or Plantation, in 1884. At that time Sibang was far outside Libreville and consisted largely of forest, some of which was exploited to produce forest products such as timber and rubber, and cleared to produce agricultural products by Europeans for international commerce for example by the Woermann company (Cheek, Harvey & Onana, 2011: 45). The Munda is the estuary that forms the eastern edge of the peninsula on which Libreville sits. Tributaries of the Munda drain the Sibang area. Beginning in 1960, the population of Libreville expanded 20-fold, and its footprint expanded. Only a small part of the original forest formerly known as Sibang survived. This part measures about 400 m × 400 m as measured on Google Earth (see further details under P. gabunensis, below) and is now entitled the ‘Sibang Arboretum’. This minute remnant of forest is probably the most visited by botanists in the whole of Gabon because it is immediately adjacent to the site of the current National Herbarium, LBV (M. Cheek, 2002, personal observation). In the unlikely although hoped-for rediscovery of Pseudohydrosme buettneri, the area of occupancy would be expected to be calculated as 4 km2 using the IUCN preferred gridcells of this size, and the extent of occurrence of the same size. If it should be found anywhere in the vicinity of Libreville it is likely to be threatened by human pressures since most of the population of Gabon is concentrated here.

The Libreville region has the highest botanical specimen collection density in Gabon, with 5359 specimens recorded in digital format. It also has the highest level of diversity of both plant species overall and of endemics (Sosef et al., 2005). The coastal forests of the Libreville area are known to be especially rich in globally restricted species (Lachenaud et al., 2013). These authors detail 19 species globally restricted to the Libreville area, of which eight have not been seen recently and which are possibly extinct. Among these is Octoknema klaineana Pierre, a rainforest tree “only collected in the immediate area of Libreville at the beginning of the 20th century, and only once since” (Gosline & Malecot, 2011). Most of the collections of this possibly extinct species of Octoknema were also, as with Pseudohydrosme buettneri, from Libreville-Sibang, and were mainly made in the period 1896–1912, during the colonial period, before the city expanded to its current extent. The other seven species recorded as globally restricted to the Libreville area and as possibly extinct by Lachenaud et al. (2013): are Ardisia pierreana Taton (Taton, 1979), Dinklageella villiersii Szlach. and Olszewski (Szlachetko & Olszewski, 2001), Eugenia librevillensis Amshoff (Amshoff, 1958), Hunteria hexaloba (Pichon) Omino (Omino, 1996), Pandanus parvicentralis Huynh (Huynh, 1986), Psychotria gaboonensis Ruhsam (Ruhsam, Govaerts & Davis, 2008) and Tristemma vestitum Jacq.-Fél. (Jacques-Félix, 1986). These species have also not been seen in several decades, or more, in the case of the penultimate species, since 1861.

The explanation for this hotspot of unique species, fast disappearing if not already extinct, at Libreville may be that it has the highest rainfall in Gabon (Gosline & Malecot, 2011), with c.2.9 m p.a.

It seems likely that Pseudohydrosme buettneri is an additional lost endemic species to the Libreville area, likely rendered extinct by the expansion of the city. Let us hope it is rediscovered in a fragment of forest in the greater Libreville area, although this seems extremely unlikely given that it was the most spectacular species of the genus with by the largest spathe (80 cm long) known in the genus, and that as stated above, the Libreville area is the most intensively botanically surveyed part of Gabon (Sosef et al., 2005).

Sect. Pseudohydrosme

Pseudohydrosme Engl. Sect. Chorianthera Engl. (Engler, 1911: 48). Homotypic synonym

Type species: Pseudohydrosme gabunensis Engl.

Male flowers in indistinct clusters of 3–5 or in pairs; distal half of spadix lacking sterile male flowers; ratio of female:male spadix portions 1:2

2. Pseudohydrosme gabunensis Engl. (Engler, 1892: 455; Engler & Prantl, 1897: 59; Brown in Thistleton-Dyer, 1901: 161; Engler, 1911: 48; Bogner, 1981: 33; Hetterscheid & Bogner, 2013: 104–113)—Figs. 3 and 5.

Figure 5 Pseudohydrosme gabunensis .

Photo of flowering, leafless plant cultivated at University of Vienna Botanic Garden (courtesy of David Prehsler).

Holotype: Gabon, Estuaire Province, Libreville, Sibang, “Gabun, Mundagebiet; Sibang-Farm am Ufer des Maveli” fl. 13 October 1881, Soyaux 299 (Holotype: B100165306, Herbarium specimen, image!)

Terrestrial herb, rhizome light brown, ellipsoid or subcylindric, erect or oblique, 9–12 cm diam. to 15 cm long, surface with transverse ridges. Roots fleshy 5–8 mm thick, brownish yellow, sometimes developing new plants at their tips (Hetterscheid & Bogner, 2013).

Leaf 1–1.3(–2.2) m tall, petiole terete, 1–1.4 cm diam. at base, dark green olive and spotted, with small yellowish white points; prickles 1–2 mm long. Blade of youngest seedlings sagittate-elliptic c. 5 cm long, 3–4 cm wide, basal sinus c. 2 cm long, breadth variable (see Hetterscheid & Bogner, 2013). Successively formed blades developing slits and divisions. Blade of mature leaves dracontoid, primary divisions 30–35 cm long, pinnatisect, lobes in each division 5–8, dimorphic, larger, distal lobes elliptic (4–)8–23 cm long, (2–)3–7(–11) cm, apex truncate, bifid, (0.5–)1–3 cm long; smaller, proximal leaflets ovate, 4.5–8 cm long, 2.5–5 cm wide, apex cuspidate; lateral veins 4–8 on each side of the midrib, conspicuous on abaxial surface, running to the margin or forming an incomplete submarginal nerve, higher order veins reticulate.

Inflorescence: Cataphylls 4–6, membranous, reddish white or brown-purple, slightly spotted, phyllotaxy spiral, proximal ones subtriangular shorter, distal ones becoming longer and oblong elliptic, towards the spathe 1.5–29 cm long, (1–)2–2.5 cm wide; peduncle (3–)5–9 cm long, 1–1.5 cm diam., colour as petiole, with minute sparse greenish white prickles 1–2 mm long. Spathe (30–)40–55(–70) cm long, fornicate, basal half (20–25 cm long) funnel-shaped to subcylindrical, fleshy and to 5 mm thick, limb comprising the distal half of the spathe, flaring widely and curving forward, the apex obtuse, margin undulate; outer surface uniform bright pale yellow, greenish yellow or yellow white; inner surface of blade mostly pale yellow or yellowish white, in an irregular marginal band, with a dark purple central area separated by an irregular margin down to the base of the tube; mouth facing horizontally, usually orbicular or elliptic. Spadix with “unpleasant smell, but not so strong as some Araceae” (Van der Laan 7641, WAG) or “faintly of lettuce” (Hetterscheid & Bogner (2013)) or “of slightly bad cabbage” (David Prehsler, University of Vienna communication to Cheek November 2020), sessile, subcylindrical, (6–)9–12.5 cm long, (1.5–)2–2.5 cm diam. Female zone (2–)3.5(–4) cm long, female flowers completely covering the surface of the axis, usually contiguous with but constricted at the junction with the male zone. Male zone (3.5–)6–8.5 cm long, at base abruptly wider than the female zone, tapering to the rounded apex, completely covered in fertile male flowers. Sterile appendix absent.

Male flowers with 2–5 stamens, stamens densely packed, sometimes paired or in groups of 3 or 5, sessile, 4 mm long, in plan view isodiametric, subprismatic, 5–6-faceted, in cross section c. 1.8(–2) mm × 1.2 mm wide, apex convex purple, sides white, anther thecae c. 3 mm long, opening by an apical pore, pollen orange or yellow, in strings. Female flowers white with ovary yellowish-white globose or ellipsoid, 2–3 mm diam., 2(–3)-locular; style 1–1.5 mm long, 1.5 mm diam., stigma black to reddish brown, surface papillose, 2 mm wide, bilobed, lobes with a broad concave area, apex rounded.

Berry, thinly fleshy, transversely ellipsoid, laterally compressed, rarely globose, 0.8–1.2 cm long, 1.5–1.6 cm wide, white, ripening purple-black, surface wrinkled when ripe, style and stigma persistent, (1–)2-seeded, apex rounded-truncate, base stipitate, stipes (2–)3–4 mm long, c. 2 mm diam. Seeds subglobose to broadly ellipsoid, one side flattened, the other convex, 9 mm long, 7 mm wide.

Phenology: flowering in the wild mid-September–late October.

Distribution and ecology: Gabon, Estuaire, Moyen-Ogooué (probably) and Woleu-Ntem Provinces, known from five sites in lowland rainforest sometimes with Aucoumea gabonensis (Burseraceae); 0–100 m alt. Possibly also in Congo (location unknown, see notes below).

Etymology: meaning “coming from Gabon” (formerly, in German “Gabun”).

Local names and uses: none known.

Additional specimens: Gabon, Woleu-Ntem Province, c. 15 km NE Asok, 600–700 m alt., (leg. Breteler and De Wilde s.n. 21 August 1978) cult. Wageningen, fl. 13 March 1984, Van der Laan 764 (Bot. Gard. No. 978PTGA550), WAG0351246, WAG0351247 images!); Estuaire Province, Libreville, Sibang: “Sibang”, hinter der Station forêstier; wächst im sandigen Lehmboden im Regenwald, sehr schattig, c. 20 m, fl. 29 October 1973, Bogner 664 (K!, M n.v. US n.v.); Sibang, st. 10 April 1994, Wieringa and Haegens 2710 (WAG0181636, WAG0181637 images!); Sibang Forest, st. 1 Dec. 1994, Wieringa 3308 (WAG0181631, WAG0181632, WAG 0181633, WAG0181634, WAG0181635 images!); Sibang Arboretum fl. 25 Oct. 2005, Sosef et al. 2029 (WAG 0223594, WAG0223595 images!, WAG8004057, WAG0108030, WAG.1665445); Kango, plantations de Assouko, près de poste de Kango, le Komo (estimated as 0° 10′ 41.8″ N, 10° 06′ 45.54″ E), fl. 2 Oct. 1912, Chevalier 26828 (P02093245 image!); Forêt de la Mondah, road from Libreville to Santa Clara, fl. 16 Sept. 1981, Breteler, Lemmens, Nzabi 7772 (WAG0449339, WAG0449339, WAG0449340 images!); St. Clara, Tussen ± 50–100 m, Linkerkant, Zij-pod naar St. Clara, sterile, no date, Breteler s.n. (WAG044938, image!); Moyen-Ogooué Province: “Congo français”. Ogooué (estimated as 0° 41′ 18″ S, 10° 13′ 55″ E), fl. 1894–95, Leroy 23 (PO2093240, PO2093241 images! two sheets).

Cultivated in Europe exact source unknown: ex Gabon, probably Sibang, fl. April 2012, leg. Bogner 3006 (BR0000019808871, image!).

Those specimens listed above which are sterile, for example Wieringa 3308 (voucher for DNA studies of the genus, see above), Wieringa and Haegens 2710, are only provisionally identified as P. gabunensis. It is possible that these specimens might belong to another species of the genus (although unlikely since these specimens were collected at Sibang Arboretum where in recent years only this species of the genus has been collected in flower). Equally they may even represent a species of the genus Anchomanes.

Conservation: Pseudohydrosme gabunensis is possibly extinct at some of its historical locations and is threatened at all of those which remain. At the type location, Sibang, formerly far outside Libreville, at least four gatherings have been made in what is now a small and highly visited forest patch inside Libreville (see notes under P. buettneri above). Measured on Google Earth, the forest is approximately a square, c. 470 m N to S and 420 m W to E, or about 0.25 km2 (grid reference: 0° 25′ 56.05″ N, 9° 29′ 23.64″ E, 49 m alt.). It is now completely surrounded by the dense urban settlement of Libreville which has expanded greatly in the last 60 years. In 1960, at independence, the population of Libreville was 32,000. Since then it has expanded 20-fold to, in 2013, 703,904 (https://en.wikipedia.org/wiki/Libreville, accessed 19 September 2020) and has a vastly greater footprint. Sibang Arboretum, the surviving patch of forest of a once much greater area, is now known as one of the top two tourist destinations in Libreville.

At the Cap Santa Clara location, the Forêt de la Mondah, known since 2012 as the Raponda Walker Arboretum (Walters et al., 2016), two collections were made, one in 1981 (see additional specimens). Since created as a protected area in 1934, it has been reduced in size, losing 40% of its area in 80 years to habitat clearance and degradation due to its close proximity (c. 15 km) to the metropolis of Libreville which draws upon its trees for timber and firewood (Walters et al., 2016). It is not clear if either of the two specimens from St. Clara were from within the current protected area.

The species has not been recorded from the Ogouué River since it was collected there by Leroy (1894–1895), despite intensive recent surveys in the lower reaches of the river whence it was probably collected. We have georeferenced the Leroy record on Lambarene since in Leroy’s time this was a trading post on the lower reaches of the river and it is credible that he stopped and collected there, but this is uncertain. The historic site on the Komo River at Kango, whence it was collected by Chevalier (26828, P; fl. 2 Oct. 1912) is now on a major transnational route, and on Google Earth shows multiple cleared areas due to development. It is possible that it no longer survives at this location, especially since it has not been recorded here or anywhere near, in a century, despite the peak decades of botanical collection in Gabon having been at the end of the 20th century (Sosef et al., 2005). Pseudohydrosme gabunensis was assessed as Endangered, EN B2ab (ii, iii) by Lovell & Cheek (2020) since it is or was known from ten specimens at five locations globally, with an area of occupation estimated as 24 km2 using the 4 km2 cell sizes preferred by IUCN (2012) and the threats detailed above. Threats in the Libreville area have already resulted in the possible global extinction of nine species, including Pseudohydrosme buettneri (see under that species, above). The extent of occurrence is calculated as 4,150 km2. If the identification of the Congolese specimen can be completely confirmed as this species, and the site of its collection discovered, the area of occupation will likely be increased to 28 km2 and the extent of occurrence also increased.

Notes. The location given in the protologue for the type specimen (see above) is similar to that of Pseudohydrosme buettneri but more detailed. The Munda is the estuary that forms the eastern edge of the peninsula on which Libreville sits. Tributaries of the Munda drain the Sibang area, one of which may have been known as the Maveli, on the forested banks of which Soyaux recorded collecting the type of Pseudohydrosme gabunensis. Herman Soyaux is reported to have collected herbarium specimens from Loango in Gabon from 1875 to 1882 (Anon, 1901).

The specimens Leroy 23 and Chevalier 26828 (both P) had been determined as Amorphophallus until identified by Bogner (M) as Pseudohydrosme gabunensis in Dec. 2012. In contrast, Wieringa 4358 (WAG) determined as this species, and cited as such in Sosef et al. (2005) is in fact an Amorphophallus, evident in the larger leaf blade divisions all being acuminate not bifid, and the tuber being described as having the roots from the top (vs.scattered along the length). Similarly, Wieringa 3308 (WAG), correctly cited in Sosef et al. (2005) as Pseudohydrosme gabunensis, was originally collected as an Anchomanes until determined by Hetterscheid in April 1996. Van der Laan 764 (WAG) had been identified as Anchomanes nigritianus Rendle until redetermined by Bogner in September 2012.

Pseudohydrosme gabunensis is the most common and widespread member of the genus. However, it is still extremely rare and with a highly restricted range in the wild. It is sought after by private collectors of aroids and live rootstocks and seed attract high prices on the internet. Fortunately, it is found in several large public botanic gardens including in Australia, Germany, France, Netherlands, UK and USA. We believe that plants are probably not collected from the wild (but this cannot be ruled out), rather they are propagated from those already in cultivation, probably from seed derived from the Netherlands.

The collection reported in Hetterscheid & Bogner (2013) as from Congo must be treated with caution. Since it is greatly disjunct (at least c. 300 km) from the known range of this species, it may even represent a further new species. We have not been able to view this specimen. However, Hay (in litt.) states that he has seen photographs of flowering material and that it looks extremely like P. gabunensis in terms of spathe shape and colouration, therefore this is the most likely identification. The specimen concerned should be located, the spadix studied carefully to determine the species beyond doubt, and an attempt made to rediscover the source population.

Differences between Pseudohydrosme gabunensis and P. buettneri are detailed under the last species. There is no doubt that Pseudohydrosme gabunensis is much more closely similar to P. ebo than to P. buettneri. However, the larger size of the spathes in P. gabunensis, their different colour and patterning, the usually bilobed style and bilocular female flowers densely covering the axis, all serve, together with the vegetative characters, to separate it from P. ebo (see also Table 1 below).

Floral visitors: Bogner (1981) collected as inferred pollinators two different flies identified as Diptera: Choridae, Sphaeroceridae, and two different beetles identified as Coleoptera: Scaphidiidae, Staphylinidae in association with Bogner 664.

Reproductive biology: Hetterscheid & Bogner (2013: 106) working with cultivated plants, report that the female flowering phase is indicated by a faint yet clear lettuce-like scent as the spathe opens, at which time, for 2 days, the receptive stigmas are wet and sticky. After this time the stigmas turn darker brown, desiccate and are no longer receptive. Individuals are obligate outcrossers. Fruits take up 10 months to mature (Hetterscheid & Bogner, 2013).

Germination and development: Germination takes 3 weeks to 10 months, producing a single small sagittate, entire leaf from a small rhizome. For several months to two years, new leaves are produced consecutively, usually each larger than its predecessor (Hetterscheid & Bogner, 2013). From the second leaf onwards slits may develop in the blade, and within two years the successively produced blades first becomes divided and finally develop the mature dracontoid pattern (see description). First flowering has occurred in as little as five years from first sowing (Hetterscheid & Bogner, 2013). In the wild, the time to maturity is likely to take longer due to predation, competition, and likely lower availability of nutrients

3. Pseudohydrosme ebo Cheek, sp. nov.—Figs. 1–3, 6 and 7.

Figure 6 Pseudohydrosme ebo (van der Burgt 1888, K, YA).

Drawing of flowering plant. ((A) ventral side view and (B) oblique side view) habit, rhizome and inflorescences; (C) inflorescence, longitudinal section, showing spadix from base to apex with sparse female flowers, partially naked axis of female zone, separated by naked axis from the densely flowered, male zone; (D) group of three stamens, ventral side view; (E) stamen; (F) stamen, transverse section; (G) female flower, entire, side view; (H) female flower, longitudinal section; (I) ovary, transverse section. Drawn by Andrew Brown.

Figure 7 Pseudohydrosme ebo (A–D from van der Burgt 2377, K, YA; E from Morgan 25, K,YA).

(A) Primary lateral division (one of three divisions) of leaf-blade, with petiole folded, behind; (B) seedling (probably in first year); (C) seedling plant (probably in third or later year); (D) seedling leaf-blade (pre-mature); ((E) obique view (B) and (F) from above); trilobed stigma. Drawn by Andrew Brown.

Differing from Pseudohydrosme gabunensis Engl. in the ovaries 3-locular, the stigma conspicuously 3-lobed, very rarely 2-locular/lobed (vs. usually 2-locular, 2-lobed, rarely 3-locular/lobed), the female zone of the spadix only sparsely covered in flowers, the spadix axis visible between the flowers (vs. completely covered in flowers), the spathe at anthesis 24–30(–34.5) cm long, the outer surface dull white with longitudinal brown stripes, inner surface light reddish brown with wide pale green veins (vs.(30–)40–55(-70) cm long, uniformly white, green or yellow on both surfaces, inner surface bicoloured, the mid-limb area dark purple, sharply demarcated from the marginal white/yellow coloured area). Type: B. J. Morgan 25 (holotype K!; isotypes B! MO! YA!), Cameroon, Littoral Region, Yabassi-Yingui, Ebo proposed National Park, fl. September 2010.

Terrestrial herb, to 1.55 m tall. Rhizome cylindric, c. 3 cm diam. obliquely erect to almost parallel to substrate surface, only upper part exposed, surface with transverse ridges (leaf scars) about 2 mm deep, 2 mm apart. Roots adventitious, thick, fleshy, c. 5 mm diam., scattered along length of rhizome, asexual reproduction not detected.

Leaf to 1.55 m tall, petiole terete, to 2 cm. diameter at base, green, inconspicuously spotted yellow, mature plants with minute, patent, extremely sparse prickles 0.5 mm long. Blade of youngest seedlings sagittate-elliptic, 5 × 2.5 cm, apex obtuse, basal sinus 1.5 × 1.5 cm, petiole 6–7 cm long. Older seedlings, in successive years with leaves developing first slits and then divisions, becoming triangular in outline with a broad basal sinus. Blade of mature leaves dracontoid, primary division 35–40 × 38–43 cm, pinnatisect, lobes 5–8, dimorphic, larger, mainly distal lobes oblong 12.5–22 × 3.8–6.5 cm, apex acuminate or truncate-bifid (biacuminate), acumen 0.8–1.5 cm long, smaller, mainly proximal lobes ovate c. 8 × 3.5 cm; lateral veins 6–11 conspicuous on abaxial surface, on each side of the midrib, uniting to form a regular looping submarginal vein 3–6 mm from the margin, higher order veins reticulate.

Inflorescence: Cataphylls 4, phyllotaxy spiral, light brown, with light green spots, membranous, successively increasing in size from proximal to distal, the outermost triangular-broadly ovate, amplexicaul 3 × 4 cm, the third in succession narrowly lanceolate-oblong, 12 × 2 cm, the fourth 18–19 × 1.5–4 cm; peduncle 3.5–4.5 × 0.6–0.7 cm, with minute, patent, extremely sparse prickles 0.5 mm long, colour as petiole. Spathe 24–30(–34.5) × 8 cm long basal 1/2–3/4 subcylindric, convolute, funnel-shaped, 1.8–4 cm wide at 2 cm above the peduncle, 6–8 cm wide at 8 cm above the peduncle, and 8–9 cm wide at 15 cm above the peduncle, the distal part (limb), half to one third of the total spathe length, flaring widely and curving forward, hood-like, shielding the spadix, the apex with a triangular acumen 3–4 × 1 cm; outer surface of both tube and limb dull white, with pale brown-red ribs running longitudinally along veins from base of tube to mouth of limb; inner surface of spathe light reddish brown, with wide pale green veins, gradually becoming slightly darker along the midline; mouth facing horizontally, transversely elliptic, 8–10 cm high, 20–25 cm wide, margin entire. Spadix sessile, cylindrical, 50–85 mm long, 10–18 mm diam. Female zone 24 mm long, 15–18 mm wide, female flowers sparsely scattered, c. 30, laxly arranged, covering only about half the surface of the spadix axis, the axis visible between the flowers, sometimes not contiguous with the male zone, the axis then naked for up to 10 mm. Male zone 37–55 mm long, 10–14 mm wide, apex rounded, completely covered in densely arranged male flowers, sterile appendix absent.

Male flowers with 2–5 stamens, sometimes paired or in groups of 3–5, stamens free, sessile, prismatic, 5 mm long, isodiametric in plan-view, 5–6 faceted, (1.5–)2 mm diam., apex convex, minutely papillate; anther thecae lateral, tetrasporangiate (Fig. 6F), oblong-elliptic, running the length of the stamen, with apical pore (Fig. 6E). Female flowers with ovary globose, 4 mm diam., 3-locular (Fig. 6I), very rarely 2-locular, style 1–1.5 mm long, 1 mm diam., stigma pale yellow, 0.5 mm thick, 2–2.25 mm wide, strongly 3-lobed (Fig. 7E), lobes with a narrow midline groove, apex rounded. Berry and seed not seen.

Distribution and ecology: Cameroon, Littoral Region, known only from three sites at one location in the Ebo forest near Yabassi-Yingui, in late secondary and intact, undisturbed lowland evergreen forest on ancient basement complex geology, rainfall c. 3 m p.a., drier season October-March; 300–400 m alt.

Conservation: Pseudohydrosme ebo is known from only one location, with three sites along a section of valley 1.3 km long and only 50–100 mature individuals in total have been seen by the collectors (second and third authors). These sites are along former logging roads which have reverted to forest (X. Van der Burgt, 2019, personal observation) as well as intact forest. In the fourteen years since 2006, botanical surveys have been made almost annually, at different seasons, over many parts of the formerly proposed National Park of Ebo. About 3000 botanical herbarium specimens have been collected, but despite the species being so spectacular in flower, with individual inflorescences lasting potentially two weeks (if they prove to be similar in phenology to those of P. gabunensis), this species has been seen nowhere else in the c. 2000 km2 of the Ebo Forest. However, much of this area has not been surveyed during the flowering season of the species, or not surveyed at all for plants. While it is likely that the species will be found at additional sites, there is no doubt that it is genuinely range restricted. Botanical surveys for conservation management in forest areas neighbouring Ebo resulting in thousands of specimens being collected and identified have failed to find any specimens of Pseudohydrosme (Cheek et al., 1996; Cable & Cheek, 1998; Cheek, Onana & Pollard, 2000; Harvey et al., 2004; Cheek et al., 2004; Cheek, Harvey & Onana, 2010; Harvey, Tchiengue & Cheek, 2010). It is possible that the species is unique to Ebo and truly localised. The area of occupation of Pseudohydrosme ebo is estimated as 4 km2 using the IUCN preferred cell-size. The extent of occurrence is the same area. In February 2020 it was discovered that moves were in place to convert the forest into two logging concessions (e.g. https://www.globalwildlife.org/blog/ebo-forest-a-stronghold-for-cameroons-wildlife/ and https://blog.resourceshark.com/cameroon-approves-logging-concession-that-will-destroy-ebo-forest-gorilla-habitat/ both accessed 19 September 2020).

This would result in logging tracks that would allow access throughout the forest allowing poachers of rare collectable plants such as Pseudohydrosme, and timber extraction would open up the canopy and remove the intact habitat in which Pseudohydrosme grows. Additionally, slash and burn agriculture often follows logging trails and would negatively impact the populations of this species. Fortunately the logging concession was suspended due to representations to the President of Cameroon on the global importance of the biodiversity of Ebo (https://www.businesswire.com/news/home/20200817005135/en/Relief-in-the-Forest-Cameroonian-Government-Backtracks-on-the-Ebo-Forest accessed 19 September 2020). However, the forest habitat of this species remains unprotected and threats of logging and conversion of the habitat to plantations remain. Pseudohydrosme ebo is therefore here assessed, on the basis of the range size given and threats stated as CR B1+2ab (iii), that is Critically Endangered.

Additional specimens: Cameroon, Littoral Region, Ebo proposed National Park, fl. 8 Oct. 2015 van der Burgt 1888 (K! YA!); ibid., st. (leaves) 9 Dec. 2019, van der Burgt 2377 (K!, MO!, P!, WAG!, YA!).

Phenology: flowering in September and early October; leaves early December; fruiting unknown.

Etymology: named as a noun in apposition for the forest of Ebo, in Cameroon’s Littoral Region, Yabassi-Yingui Prefecture, to which this spectacular species is globally restricted on current evidence.

Local names and uses: none are known.

Notes: Pseudohydrosme ebo came to the attention of the first author in late Aug. 2018 on seeing van der Burgt 1888, collected in 2015. Plans were made to revisit the collection site at the next available opportunity, in December 2019, when leaves were found by the third author (van der Burgt 2377), but unfortunately fruits were not found. At the same time a second site was discovered 1.3 km distant from the site found in 2015. In February 2020 van der Burgt found at Kew an overlooked, additional specimen, Morgan 25, which is the earliest known record of the species, dating from 2010, and since it has multiple duplicates, has been selected as type of the new species. The associated collection data previously mislaid was rediscovered in May 2020.

Alvarez with van der Burgt, and Ngansop, discovered in December 2019 seedlings of the new species, at three different stages, preserved as Van der Burgt 2377 sheet 1/4 (see Fig 7). Clearly the species at this site is reproducing itself. Associated photographs also show plants of different ages.

Abwe & Morgan (2008) and Cheek et al. (2018b) characterise the Ebo forest, and give overviews of habitats, species and importance for conservation. Fifty-two globally threatened plant species are currently listed from Ebo on the IUCN Red List website and the number is set to rise rapidly. The discovery of a new species to science at the Ebo forest is not unusual. Since numerous new species have been published from Ebo in recent years. Examples of other species that, like Pseudohydrosme ebo appear to be strictly endemic to Ebo on current evidence are: Ardisia ebo Cheek (Cheek & Xanthos, 2012), Crateranthus cameroonensis Cheek and Prance (Prance & Jongkind, 2015), Gilbertiodendron ebo Burgt and Mackinder (Van der Burgt et al., 2015), Inversodicraea ebo Cheek (Cheek et al., 2017), Kupeantha ebo M. Alvarez and Cheek (Cheek et al., 2018a), Palisota ebo Cheek (Cheek et al., 2018b).

Further species described from Ebo have also been found further west, in the Cameroon Highlands, particularly at Mt Kupe and the Bakossi Mts (Cheek et al., 2004). Examples are Myrianthus fosi Cheek (Cheek & Osborne, 2010), Salacia nigra Cheek (Gosline, Cheek & Kami, 2014), Talbotiella ebo Mackinder and Wieringa (Mackinder, Wieringa & Van der Burgt, 2010)

Additionally, several species formerly thought endemic to Mt Kupe have subsequently been found at Ebo, for example Coffea montekupensis Stoff. (Stoffelen et al., 1997), Costus kupensis Maas and H. Maas (Van de Kamer et al., 2016), Microcos magnifica Cheek (Cheek, 2017), and Uvariopsis submontana Kenfack, Gosline and Gereau (Kenfack et al., 2003).

Therefore, it is possible that Pseudohydrosme ebo might yet also be found in the Cameroon highlands, for example at Mt Kupe, further extending westward the known range of the genus. However, this is thought to be only a relatively small possibility given the spectacular nature of this plant, and the high level of survey effort at for example Mt Kupe: if it occurred there it is highly likely that it would have been recorded already.

Additional characters separating Pseudohydrosme ebo from P. gabunensis are show in Table 1.

It is to be hoped that further studies of live plants of P. ebo will be possible to determine if, like P. gabunensis it also reproduces asexually from the root tips.

The biogeography of the Cameroonian Pseudohydrosme ebo is very different from that of the two Gabonese species of the genus growing c.450 km to the South. The Gabonese species grow on recently deposited, sandy coastal soils. Although the Gabonese species also experience a wet season of about 3 m of rainfall per annum, it is differently distributed: the dry season in Libreville occurs from June to September inclusive and is colder than the wet season. In contrast at Ebo the geology at the Pseudohydrosme location is ancient, highly weathered basement complex, with some ferralitic areas in foothill areas which are inland, c. 100 km from the coast. The wet season (successive months with cumulative rainfall >100 mm) is almost the inverse of at Libreville, falling between March and November and is colder than the dry season (Abwe & Morgan, 2008). In addition, the affinities of Ebo as indicated by shared plant species, seems to be with other parts of the Cross-Sanaga biogeographic area, the Cameroon Highlands, rather than with Gabon (see above).

Discussion

The description of Pseudohydrosme ebo (this article) necessitates the modification of the circumscription of Pseudohydrosme in two respects. Firstly, in P. ebo the flower-bearing male and female portions of the inflorescence are not completely contiguous, the distal region of the female zone shows naked parts of the axis and the female flowers are laxly arranged throughout, while in the other species of Pseudohydrosme there is no naked portion and the spadix axis is completely covered in flowers. Thirdly the trilocular ovaries normal in Pseudohydrosme ebo are different to those of the other two species which are usually bilocular, and only very rarely otherwise.

Although indicated as potentially congeneric with Anchomanes by Hetterscheid & Bogner (2013) who cited only the difference in ovary locularity as a basis for maintaining the separation, in fact five other characters support maintaining the separation of these two genera (see Table 2 above). Two of these characters were discovered for the first time by Hetterscheid & Bogner (2013). These are (1) the development in the fruit of a stipe and (2) reproducing asexually from the fleshy roots: producing new plants distant from the parent rhizome. The last character is specifically remarked to be definitively absent from Anchomanes species, which have been studied in detail in cultivation (Hetterscheid & Bogner, 2013). However, Hay (in litt.) reports anecdotally, that Anchomanes can be propagated in cultivation by root cuttings.

Table 2 Characters separating Anchomanes and Pseudohydrosme.

	Anchomanes	Pseudohydrosme	
Spadix: spathe proportions and presentation	Spadix relatively long >1/2–9/10 as long as spathe; conspicuous, projected above the (short) spathe tube	Spadix relatively short, 1/10–1/4 as long as spathe; completely concealed within, and at base of the (long) spathe tube	
Peduncle	Peduncle exserted far beyond cataphylls; 2–5 x length of spathe	Completely concealed at anthesis by cataphylls;
<1/10 length of spathe	
Ovary locularity and stigma lobe number	1	2 or 3	
Berries	Sessile	Stipitate (where known)	
Laticifers	Present, simple, articulated	Absent (not detected by Keating, 2002)	
Reproducing asexually from roots	Absent (but propagation from root cuttings in cultivation reported anecdotally)	Present	
Note:

Data from Mayo, Bogner & Boyce (1997); Bogner (1981); Cusimano et al. (2011); Hetterscheid & Bogner (2013); this article.

However, there is no doubt that Anchomanes and Pseudohydrosme are closely related, sharing so many characters. Although the two genera have been stated to have “at least a sister group relationship” (Hetterscheid & Bogner, 2013), as those authors pointed out, only one species of each genus was included in the molecular phylogenetic analysis upon which this statement was based (Cabrera et al., 2008). We speculate that the differences in inflorescence structure that help distinguish between the two genera (Table 2), might be due to two different pollination syndromes being in play. That laticifers were not detected in Pseudohydrosme might have been due to insufficient material being available. Resampling might yet discover them. The difference in peduncle character state that currently separates the two genera are encompassed within one genus elsewhere in the family, with intermediates connecting the two states, for example in Amorphophallus. It cannot be ruled out therefore that one of these genera might be nested inside the other, in which case it might be necessary to sink Pseudohydrosme into Anchomanes. However further molecular analysis, sampling additional taxa of each genus, is advisable before any such action is needed. It is intended to address this in a future project.

Conclusions

The discovery of a new species of Pseudohydrosme in Cameroon, far from the border with Gabon, is completely unexpected after nearly 130 years in which no additional taxa have been added to the genus. It is also unexpected because one would not predict from the pre-existing data on the genus that such a new species would be so biogeographically and climatically disjunct from its congeners in the Libreville area of Gabon (see under Pseudohydrosme ebo above). However, examples of even more dramatically unexpected African range extensions have occurred recently such as the westward extension by 2,400 km of the genus Ternstroemia Mutis ex L.f., of Talbotiella Baker by 1,400 km, and of the genus Metarungia Baden by 1,200 km, or in the other direction, eastwards, 1,500 km in Mischogyne Exell (Cheek et al., 2019; Van der Burgt et al., 2018; Darbyshire, Vollesen & Chapman, 2008; Gosline, Marshall & Larridon, 2019 respectively). Such discoveries underline how incomplete our knowledge of the geography of African plant genera remains. They also underline the urgency for uncovering such taxa whileit is still possible since in all but one of the cases given, the range extension resulted from finding a new species to science with a narrow geographic range and/or very few individuals, and which face threats to their natural habitat, putting these species at high risk of extinction. About 2,000 new species of vascular plants have been discovered each year for the last decade or more. Until species are known to science, they cannot be assessed for their conservation status and the possibility of protecting them is reduced (Cheek et al., 2020). Documented extinctions of plant species are increasing, for example Oxygyne triandra Schltr. of Southwest Region, Cameroon is now known to be globally extinct (Cheek et al., 2018c). In some cases species appear to be extinct even before they are known to science, such as Vepris bali Cheek, also from the Cross-Sanaga interval in Cameroon (Cheek, Gosline & Onana, 2018) and elsewhere, Nepenthes maximoides Cheek (King & Cheek, 2020). Most of the >800 Cameroonian species in the Red Data Book for the plants of Cameroon are threatened with extinction due to habitat clearance or degradation, especially of forest for small-holder and plantation agriculture following logging (Onana & Cheek, 2011). Efforts are now being made to delimit the highest priority areas in Cameroon for plant conservation as Tropical Important Plant Areas (TIPAs) using the revised IPA criteria set out in Darbyshire et al. (2017). This is intended to help avoid the global extinction of additional endemic species such as Pseudohydrosme ebo which will be included in the proposed Ebo Forest IPA.

Ekwoge Abwe and Bethan Morgan of San Diego Zoo Global and their team at the Ebo Forest Research Project are thanked hugely for expediting our botanical surveys in the Ebo forest of Cameroon over several years. In particular Bethan Morgan is acknowledged for collecting the type specimens of Pseudohydrosme ebo.

Janis Shillito is thanked for typing the manuscript. The heads of IRAD (Institute of Research in Agronomic Development)-National Herbarium of Cameroon, Yaoundé, successively Jean-Michel Onana, Florence Ngo Ngwe and Eric Nana, are thanked for arranging permits and co-ordinating the co-operation with the Royal Botanic Gardens, Kew. The late Josef Bogner is thanked for conversations on Araceae. Maria Alvarez is thanked for photos of seedlings of Pseudohydrosme ebo. Marcello Sellaro of the Tropical Nursery, Royal Botanic Gardens, Kew, is thanked for cultivation of and facilitating access to live material of Araceae. David Prehsler, University of Vienna Botanic Garden is thanked for the photo of Pseudohydrosme gabunensis and for notes on the scent of that plant. Alistair Hay, Simon Mayo and another, anonymous reviewer, are all thanked for their detailed, expert and painstaking reviews of an earlier version of this paper, which have greatly improved the final version.

Additional Information and Declarations

Competing Interests

Author Contributions

Ethics

Field Study Permissions

Data Availability

New Species Registration

The authors declare that they have no competing interests.

Martin Cheek conceived and designed the experiments, performed the experiments, analyzed the data, prepared figures and/or tables, authored or reviewed drafts of the paper, and approved the final draft.

Barthélemy Tchiengué performed the experiments, authored or reviewed drafts of the paper, field studies, and approved the final draft.

Xander van der Burgt performed the experiments, prepared figures and/or tables, authored or reviewed drafts of the paper, field studies, and approved the final draft.

The following information was supplied relating to ethical approvals (i.e., approving body and any reference numbers):

The fieldwork was approved by the Institutional Review Board of the Royal Botanic Gardens, Kew entitled the Overseas Fieldwork Committee (OFC) under registration numbers OFC 673-1 (2015) and OFC 807-3 (2019).

The following information was supplied relating to field study approvals (i.e., approving body and any reference numbers):

IRAD-Herbier National du Cameroun sanctioned the field work under a series of Memoranda of Collaboration with the Royal Botanic Gardens, Kew, the most recent signed 4 September 2019, extending to 5th September 2021 (Research permit number: 000146/MINRESI/B00/C00/C10/C12).

The following information was supplied regarding data availability:

All raw data is available in the Results section of the article.

All specimens and associated metadata are also cited within the article’s Results section, either as type specimens or under “additional specimens”. Additionally, we also duplicate this information here for easy reader access:

- Pseudohydrosme buettneri Engl.

Holotype: Gabon, Estuaire Province, Libreville “Gabun, Mundagebiet; Sibange-Farm” fl. Sept. 1884, Buettner 519 (Holotype B, destroyed or mislaid).

- Pseudohydrosme gabunensis Engl.

Holotype: Gabon, Estuaire Province, Libreville, Sibang, “Gabun, Mundagebiet; Sibang-Farm am Ufer des Maveli” fl. 13 October 1881, Soyaux 299 (Holotype: B100165306, image!)

Additional specimens: Gabon, Woleu-Ntem Province, c. 15 km NE Asok, 600–700 m alt., (leg. Breteler and De Wilde s.n. 21 Aug. 1978) cult. Wageningen, fl. 13 March 1984, van der Laan 764 (Bot. Gard. No. 978PTGA550), WAG0351246, WAG0351247 images!); Estuaire Province, Libreville, Sibang: “Sibang”, hinter der Station forêstier; wächst im sandigen Lehmboden im Regenwald, sehr schattig, c. 20 m, fl. 29 Ocktober 1973, Bogner 664 (K!, M n.v. US n.v.); Sibang, st. 10 April 1994, Wieringa and Haegens 2710 (WAG0181636, WAG0181637 images!); Sibang Forest, st. 1 Dec. 1994, Wieringa 3308 (WAG0181631, WAG0181632, WAG 0181633, WAG0181634, WAG0181635 images!); Sibang Arboretum fl. 25 Oct. 2005, Sosef et al. 2029 (WAG 0223594, WAG0223595 images!, WAG8004057, WAG0108030, WAG.1665445); Kango, plantations de Assouko, près de poste de Kango, le Komo (estimated as 0° 10′ 41.8″ N, 10° 06′ 45.54″ E), fl. 2 Oct. 1912, Chevalier 26828 (P02093245 image!); Forêt de la Mondah, road from Libreville to Santa Clara, fl. 16 Sept. 1981, Breteler, Lemmens, Nzabi 7772 (WAG0449339, WAG0449339, WAG0449340 images!); St. Clara, Tussen ± 50–100 m, Linkerkant, Zij-pod naar St. Clara, sterile, no date, Breteler s.n. (WAG044938, image!); Moyen-Ogooué Province: “Congo français”. Ogooué (estimated as 0° 41′ 18″ S, 10° 13′ 55″ E), fl. 1894–95, Leroy 23 (PO2093240, PO2093241 images! two sheets). Cultivated in Europe exact source unknown: ex Gabon, probably Sibang, fl. April 2012, leg. Bogner 3006 (BR0000019808871, image!).

Pseudohydrosme ebo Cheek, sp. nov.

Type: B. J. Morgan 25 (holotype K001381516!; isotypes B! MO! YA!), Cameroon, Littoral Region, Yabassi-Yingui, Ebo proposed National Park, fl. September 2010.

Additional specimens: Cameroon, Littoral Region, Ebo proposed National Park, fl. 8 October 2015 van der Burgt 1888 (K001286047! YA!); ibid., st. (leaves) 9 Dec. 2019 van der Burgt 2377 (K001381468, K001381469! K001381470!, K001381471! MO!, P!, WAG!, YA!).

All specimens are herbarium specimens.

The following information was supplied regarding the registration of a newly described species:

Pseudohydrosme ebo 77213519-1.

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
