# Peer review of "Taxonomic revision of the threatened African genus Pseudohydrosme Engl. (Araceae), with P. ebo, a new, critically endangered species from Ebo, Cameroon"

_PeerJ, doi:10.7717/peerj.10689_

## Round 0.1 · original submission · Major Revisions

Both reviewers coincided that it is necessary to provide more arguments to justify Pseudohydrosme and Anchomanes as independent. In addition, more detail is needed in methods to understand how the characters were gathered. Also, one of the reviewers suggested a number of issues in the Table of attributes and to provide a summary for the diagnostic characters at the level of genus.

·

Basic reporting

No comment beyond minor matters in the attached line-by-line feedback.

Experimental design

The paper is not experimental, so no comment.

Validity of the findings

No comment

Additional comments

The discovery of Pseudohydrosme ebo is an important find and a significant range extension for this highly endangered and very small tropical West African genus. The new species is clearly very closely related to, but sufficiently morphologically and biogeographically distinct from, Pseudohydrosme gabunensis. The contrast drawn between the leaflet venation of P. gabunensis and P. ebo appears to be overstated, but the species are nevertheless distinguishable on inflorescence characteristics, supported by the marked geographic disjunction. Pseudohydrosme itself is *extremely* closely allied to the genus Anchomanes, and likely to eventually be subsumed into it. The authors have taken a prudent, conservative approach by describing the new species in Pseudohydrosme, but it would also be open to them to describe the new species in, and transfer the already recognised two species into, Anchomanes, though that could perhaps await a better sampled molecular analysis than is currently available. I think this should be acknowledged in the paper. One fairly significant matter is that the authors stress that Pseudohydrosme and Anchomales are sister genera, but this has not actually been demonstrated in the molecular phylogenetic analyses cited, wherein each genus has been represented by only one species with the consequence that it is impossible to discern if Pseudohydrosme is sister to, or nested within, Anchomanes. This is rather cryptically acknowledged in the paper, yet conflicts with the bald statement that the genera have a sister relationship. I have itemised some further minor errors and quibbles in the attached line-by-line review.

Reviewer 2 ·

Basic reporting

Authors are reporting a new species P. ebo. Authors provide all the detail very effectively and the study is reproducible in all aspect. However, authors did not performed some important analyses.

Experimental design

The study is well explained and reproducible up to the data authors provided in the manuscript. However, I have some concerns as authors did not perform pollen and molecular related studies.

Validity of the findings

The addition of pollen grain study and molecular markers can make the study completely valid.

Additional comments

Dear authors,
I appreciate your efforts and you have provided a lot of detail.
I have some suggestions.
1. You have mentioned the use of microscope in methodology but the reader can not get your point from methodology that which kind of characters you examine. Please improve it.
2. You need to provide complete detail of the characters of the three species of pseudohydrosme in a table instead of mentioning 2 species only. This will be helpful to provide all relevant information to the readers. Please shift details related to the genus pseudohydrosme, its sections, and of two other species into supplementary. I think these information can make the reader confusing. You need to just provide a short summary of the characters of the genus and its section as well as each species instead of providing the details which is already well documented. Then, you can provide complete detail of P. ebo. You can also follow Cusimano et al. and can provide a list of character as descriptors which you have focused on pseudohydrosme and its related genera in the methodology. This will be good to make the methodology further clearer.
3. I think you have not provide data about the morphology of pollen. I think the electron microscopy of the pollen analyses will be great to get further insight into the similarities and variations among the species of pseudohydrosme.
4. You can see the molecular data is already reported for the genera of Araceae. You have already cited those article therefor I am not going to mentioned those articles. I think the generation of the molecular data of even few loci will be better to provide molecular level evidence. Please avoid the used of trnH-psbA and matK as the recent study show the shifting of these genes from LSC to IR in Anchomanes. Moreover, the recent study also show the effect of rate Heterotachy.
https://www.sciencedirect.com/science/article/pii/S0888754319308420
https://link.springer.com/article/10.1007/s00239-020-09958-w
The phylogeny can be constructed in comparison to previous data. You just need to sequence 3-4 loci per species for only species of pseudohydrosme.
5. Please improve the last lines of discussions. You are providing some suggestions but I feel the language can be improved by rewriting and by shuffling information of those lines.

---

## Round 0.2 · accepted · Accept

Thank you for considering previous reviews, Dr. Hay only pointed out some typos that need to be corrected during the editorial process, they are listed below.

·

Basic reporting

No comment

Experimental design

no comment

Validity of the findings

no comment

Additional comments

Referees comments and corrections have been appropriately addressed, and I recommend acceptance subject to correction of some typos (see below)

The author should check that all references in the body of text are italicised and include a comma before the date: there are some inconsistencies.

I draw particular attention to a typo in line 576 where the population increase is surely 20-fold not 200-fold as stated!

Line 216: African
258: delete 2nd comma after fornicate, and slash before very
407: full stop after clade
438: Sasaf et al., 2005
497: comma after 4–6
516: 2–5 [i.e. en-dash not hyphen]
518: 3 mm [i.e. add space]
530: en-dash, not hyphen, after Sept.
576: 20, not 200
670: locular
699: fonts for half and three quarters are different
707: comma after cylindrical, and close up 50 and en-dash
714: 2–5 [i.e. en-dash not hyphen]
717: pores